# Brain Abscess Masquerading as Brain Infarction

**DOI:** 10.3390/brainsci10070440

**Published:** 2020-07-11

**Authors:** Da-Eun Jeong, Jun Lee

**Affiliations:** 1Department of Neurology, Veterans Health Service Medical Center, 53 Jinhwangdo-ro 61-gil, Gangdong-gu, Seoul 05368, Korea; doctorjung86@gmail.com; 2Department of Neurology, Yeungnam University Medical Center, 170 Hyeonchung-ro, Namku, Daegu 42415, Korea

**Keywords:** brain abscess, brain infarction, embolism, brain, pneumonia, bacterial, diffusion magnetic resonance imaging

## Abstract

Occasionally, acute ischemic stroke can be difficult to differentiate from acute intracranial infection. We describe a patient who presented with sudden onset of right hemiparesis and fever. Magnetic resonance imaging (MRI) was consistent with an acute stroke, showing multiple lesions with restricted diffusion in the left middle cerebral artery territory. These lesions were not enhancing and were not associated with vasogenic edema. A diagnosis of acute stroke was made based on the clinical and radiographic data. Follow-up MRI obtained eleven days later showed interval development of ring enhancement and vasogenic edema surrounding the previously noted core of restricted diffusion. Based on these findings, the diagnosis was revised to cerebral abscesses and the patient was treated successfully with antibiotics. In retrospect, the largest diffusion-weighted lesion on baseline MRI demonstrated two characteristics that were atypical for stroke: it had an ovoid shape and a subtle T2 hypointense core. This case demonstrates that acute clinical and radiographic presentation of cerebral abscess and ischemic stroke can be strikingly similar. Follow-up imaging can be instrumental in arriving at an accurate diagnosis.

## 1. Introduction

Brain abscesses typically present with progressive headache, an altered level of consciousness, progressive focal neurologic deficits, and/or seizure [1]. Occasionally, patients can present with sudden onset focal neurological symptoms and it can be difficult to differentiate cerebral abscess from acute ischemic stroke on clinical grounds. Differentiating these two entities based on radiological findings is usually straightforward but can sometimes be challenging as ischemic infarcts and cerebritis/early cerebral abscesses can have overlapping magnetic resonance imaging (MRI) findings in the acute setting. To illustrate this potential pitfall, we present the case of a patient with a brain abscess who was initially misdiagnosed as having acute cerebral infarction based on his clinical and imaging presentation.

## 2. Case Presentation

A 65-year-old man with a past medical history of diabetes mellitus developed sudden weakness of the right arm and leg. He presented the following day to our hospital; on physical examination, he was found to have a fever of 38.7 °C, mild right hemiparesis (Medical Research Council scale grade 4), right facial palsy, and dysarthria. He scored four points on the National Institutes of Health Stroke Scale. His cardiac examination findings, including echocardiography (EKG) and transthoracic echocardiography, were unremarkable. MRI of the brain was performed and multiple high signal intensity lesions were seen in the left frontal and parietal lobes on T2-weighted MRI and on diffusion-weighted imaging (DWI) with an associated low signal on the apparent diffusion coefficient (ADC) map (Figure 1). We subsequently performed contrast-enhanced brain MRI and magnetic resonance angiography (MRA). There was no evidence of enhancement of the lesions on post-gadolinium images and there was no evidence of hemorrhage on T2 *-weighted images. The lesions were all felt to be in the left middle cerebral artery territory (although there was some question as to whether the most inferior lesion was in the posterior cerebral artery territory). However, contrast-enhanced brain MRA showed no steno-occlusion of the left middle cerebral artery. An initial diagnosis of acute cerebral infarction was made on the basis of these findings and the patient was treated with antiplatelet therapy. Because of the patient’s fever, an infectious work-up was performed and blood cultures were drawn. The complete blood count was unremarkable but chest X-ray suggested bronchitis. The patient was empirically treated with netilmicin and cefmetazole and his fever resolved within 2 days of admission to the hospital. On day five, Enterobacter aerogenes was found in his blood cultures.

Over the next several days, the patient did well and was presumed to have experienced an embolic stroke as well as bronchitis complicated by septicemia. On the eleventh day after symptom onset, the patient experienced a focal motor seizure involving his right hand and follow-up MRI of the brain was performed. The T2-weighted MRI was notable for interval development of extensive vasogenic edema surrounding the left-sided brain lesions and the post-gadolinium T1-weighted MRI images demonstrated the ring enhancement of these lesions (Figure 2). Based on these findings, the patient’s diagnosis was revised to multiple brain abscesses and he was switched to a 6-week course of triple antibiotic therapy with ceftriaxone, vancomycin, and metronidazole. The patient improved clinically and was discharged home without residual deficits after two months. Follow-up MRI at this time showed a marked decrease in the size of the lesions and associated edema (Figure 3).

## 3. Discussion

This case illustrates that the clinical and radiographic presentation of a patient with evolving cerebral abscesses can be confused with cerebral infarction. Cerebral abscesses can present with acute focal “stroke-like” symptoms and the MRI may demonstrate cerebral lesions with restricted diffusion and no contrast enhancement, mimicking acute infarcts. Although cerebral abscesses are rare compared with ischemic infarcts, knowledge of the potential similarities in both the clinical and imaging presentation of these disorders can facilitate accurate diagnosis and prompt initiation of appropriate treatment.

The typical symptoms of brain abscess include slowly progressive headache with an alteration in level of consciousness and neurological deficits. This case illustrates that some patients, however, present with acute stroke-like symptomatology. This is consistent with previously reported cases of an acute stroke-like presentation of cerebral abscesses [2,3,4,5]. Although the exact mechanism of stroke-like onset of symptoms in bacterial brain abscess is unknown, paroxysmal septic emboli or primary cerebral infarction associated with systemic bacteremia are considered possible mechanisms [3]. In older reports, brain imaging was generally limited to CT and the authors suggested that misdiagnosis of cerebral abscesses may be avoided by obtaining acute MRI scans [3,5]. Our case demonstrates that misdiagnosis can occur even when high-quality MRI is obtained. Several MRI characteristics can lead to difficulty in differentiating abscesses from acute strokes. First, high signal on DWI and low signal on the corresponding ADC map are imaging hallmarks of acute cerebral ischemia. These findings are, however, not pathognomonic for acute stroke and can also be seen in brain abscesses, acute demyelinating disease, repetitive seizures, and brain tumors [6,7] among other entities. The high DWI signal from brain abscesses results from the high viscosity of the pyogenic exudate, which impedes water diffusion [8]. The DWI signal abnormality of a brain abscess is typically confined to the central region of the abscess, but as in this case, cerebritis with nascent abscesses may result in an atypical pattern on DWI, with more homogeneous diffusion abnormality in focal or multifocal regions of brain tissue, as is seen with acute ischemic infarction. Second, contrast enhancement is typically seen with cerebral abscesses but not in acute stroke. As this case illustrates, contrast enhancement is not a universal finding in cerebral abscesses. Although contrast enhancement was present on the follow-up MRI obtained on day 11, it was not present on the baseline MRI. The lack of contrast leakage in the acute setting reflects the relative integrity of the blood–brain barrier at this stage. It is possible that subtle enhancement could have been detected with a larger contrast dose or a longer delay-time between contrast administration and post-contrast MRI, but these maneuvers are not typically performed in the routine clinical setting. Third, cerebral abscesses are generally associated with significant vasogenic edema, whereas acute strokes are not. While the follow-up MRI demonstrated the presence of extensive edema, there was no evidence of vasogenic edema on the baseline scan. This too is likely a reflection of the relative integrity of the blood–brain barrier early on in the development of the brain abscesses. Finally, the susceptibility artifact on T2 *-weighted MRI, reflecting hemorrhage, has often been described in the setting of cerebral abscesses. This same finding can be seen in acute brain infarcts and, as this case illustrates, is not always present in cerebral abscess. 

Despite the striking similarities that can exist between evolving cerebral abscesses and acute cerebral infarcts, as illustrated by our case, there are a few subtle radiographic findings that might, in retrospect, have helped to establish the correct diagnosis. First, the dominant left parietal lesion (see magnified view in the last column of Figure 1) has subtle low signal intensity in its center on the T2-weighted image; a low signal on T2 is often a feature of CNS infection. Second, a number of the lesions have an ovoid shape, which is not typical of routine embolic strokes. Clinicians should consider the possibility of septic emboli in patients with a sudden onset of neurologic deficits in the setting of fever, even if cardiologic examination shows no evidence of bacterial endocarditis, and certainly if there are stroke-like imaging findings on acute MRI. Our case underscores several important teaching points: (1) obtaining a short-interval follow-up MRI is important in patients who might have septic embolism, especially when the initial MRI and clinical features are not entirely compatible with acute bland embolic stroke; (2) the possibility of early cerebritis should be considered in a febrile patient who has lesions on MRI that have atypical features for ischemia, including low signal intensity in the center of the lesion on T2-weighted images and/or round or ovoid lesions; and (3) if abscess is in the differential diagnosis and no contrast enhancement is detected on routine imaging, increasing the gadolinium dose and/or the time delay between gadolinium injection and imaging may detect more subtle abnormalities of the blood–brain barrier.

## Figures and Tables

**Figure 1 brainsci-10-00440-f001:**
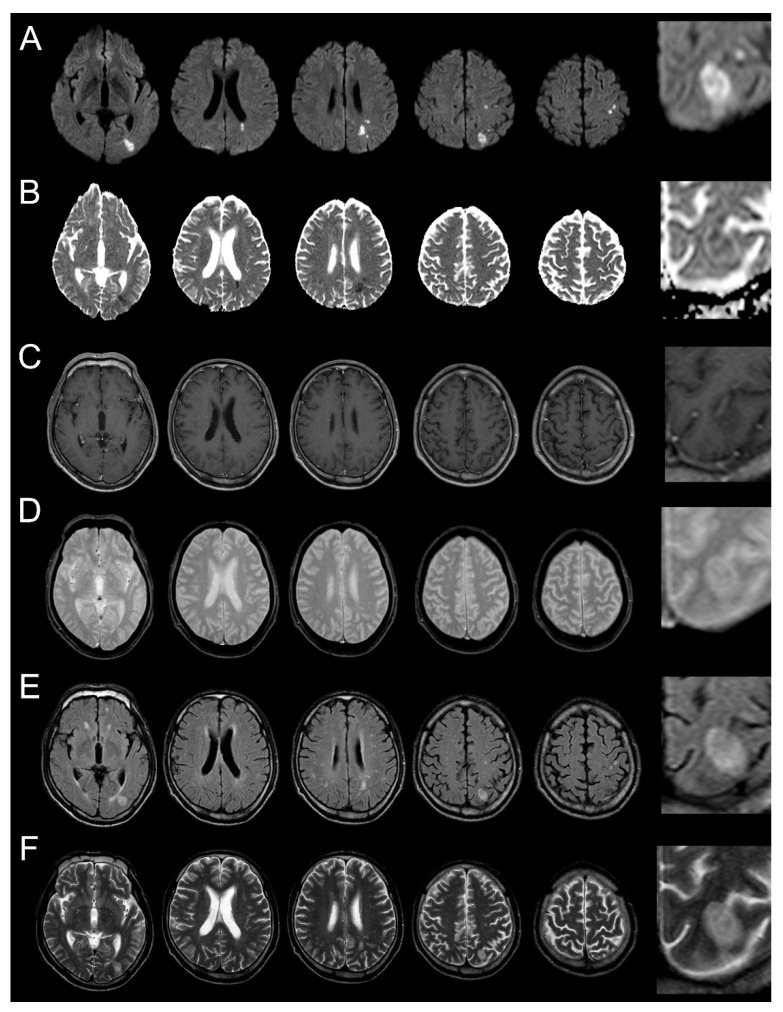
Brain MRI was performed 2 days after symptom onset. (**A**): diffusion-weighted magnetic resonance imaging (MRI); (**B**): apparent diffusion coefficient map (ADC); (**C**): contrast enhanced T1-weighted MRI; (**D**): T2 *-weighted MRI; (**E**): fluid-attenuated inversion recovery (FLAIR); (**F**): T2-weighted MRI). Multiple lesions can be seen in the border area between the middle cerebral artery and posterior cerebral artery in the left hemisphere. High signal intensity lesions on diffusion-weighted MRI and their corresponding low-signal lesions on the ADC, without enhancement on contrast-enhanced T1-weighted MRI and without low signal on T2 *-weighted MRI, are seen. T2-weighted MRI shows a high signal lesion with subtle low signal in the center of the ovoid-shaped dominant peripheral lesion. Figures in the right column are magnified regions of interest from the images of the lesion marked (*).

**Figure 2 brainsci-10-00440-f002:**
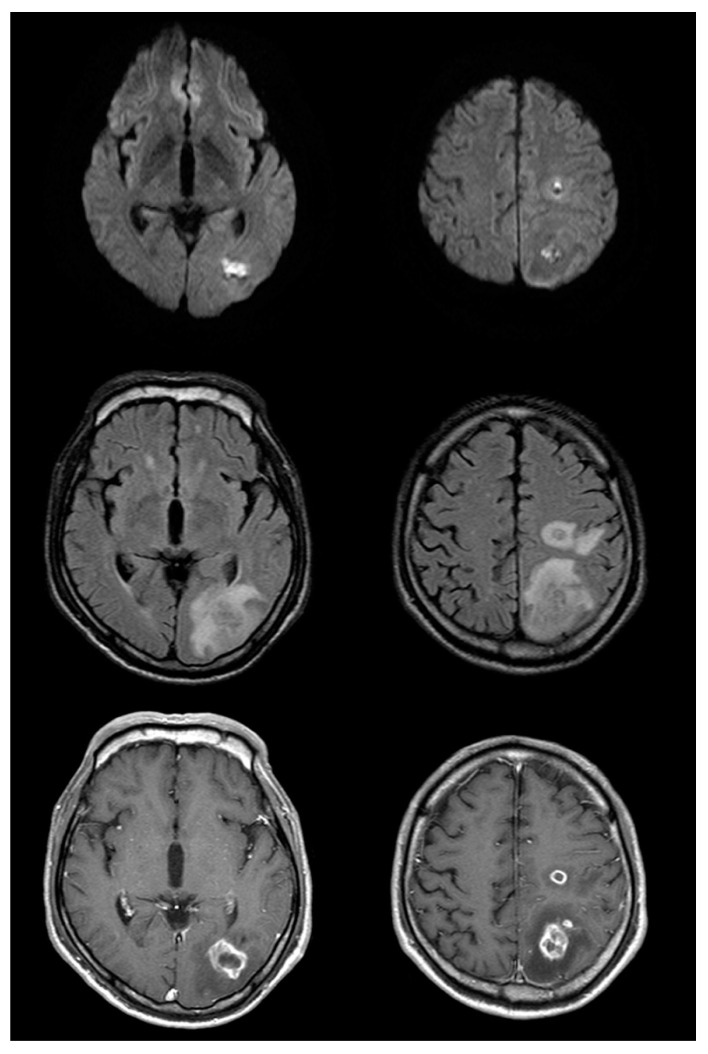
Follow up brain magnetic resonance imaging on the eleventh day after symptom onset (top row: diffusion-weighted imaging (DWI); middle row: fluid-attenuated inversion recovery (FLAIR); bottom row: post-gadolinium). There is significant interval change with enlargement of the lesions on DWI and FLAIR images, as well as significant vasogenic edema. Post-gadolinium T1-weighted images show somewhat thick and irregular but complete ring enhancement.

**Figure 3 brainsci-10-00440-f003:**
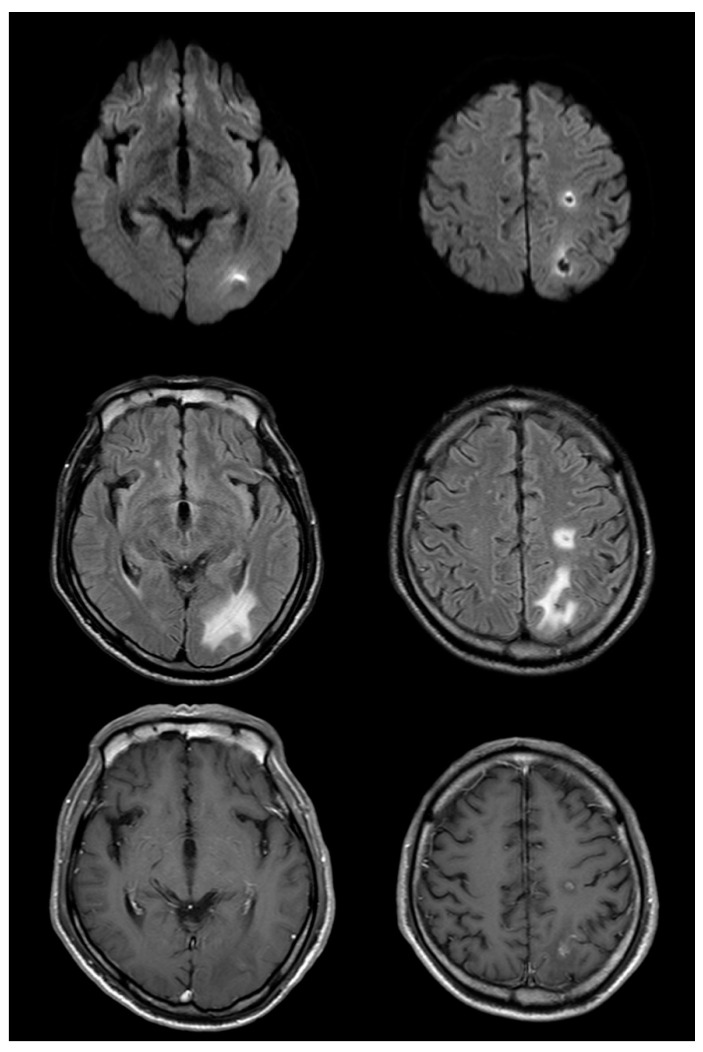
Magnetic resonance imaging (MRI) on the 52nd day after symptom onset reveals a decrease in the size of lesions compared with that in the prior study (top row: diffusion-weighted imaging (DWI); middle row: fluid-attenuated inversion recovery (FLAIR); bottom row: post-gadolinium). Edema and enhancement are markedly decreased compared to that in the previous MRI; changes originally seen on DWI are evolving and resolving as well.

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
