# Peer review of "Brain Abscess Masquerading as Brain Infarction"

_brainsci, 2020, doi:10.3390/brainsci10070440_

Round 1

Reviewer 1 Report

The authors present a case of suspected acute stroke based on neuroradiological findings, which was instead a brain abscess. Even if the report is well described, i think that the initial managment of the case led to a delay of diagnosis. Some concerns:

-despite the acute onset of right hypostenia, leading to the diagnosis of stroke, the author did non performed (or, otherwise, is not reported) angioCT examination, that could clarify the preserved flow in MCA. In fact, as author correctly assess, the hyperintensity is not clealry delimited in MCA territory, involving also other arteries.

-the presence of fever, along with predisposing condition as dyabetes, and the DWI hyperintensity with low ADC, seems more compatible with an infection rather than with a stroke. Often, in brain abscess, blood count examination is only moderately alterated.

This managment led to a delay of correct tratment, which started 11-12 days after the symtomps onset 

Author Response

The revised text and author to respond reviewer 1 are in the attached file 

Reviewer 2 Report

The case report is well written; easy to read, well constructed and of appropriate length, content and background. Clinical implications of case are well explained. I didn’t have any changes or suggestions for authors.

Author Response

The reviewer said, I didn’t have any changes or suggestions for authors.

Round 2

Reviewer 1 Report

Now the manuscript is most suitable for publication